# Diagnostic Potential of Exosomal microRNAs in Colorectal Cancer

**DOI:** 10.3390/diagnostics12061413

**Published:** 2022-06-08

**Authors:** Jonas Dohmen, Alexander Semaan, Makbule Kobilay, Martin Zaleski, Vittorio Branchi, Anja Schlierf, Karina Hettwer, Steffen Uhlig, Gunther Hartmann, Jörg C. Kalff, Hanno Matthaei, Philipp Lingohr, Stefan Holdenrieder

**Affiliations:** 1Department of General, Visceral, Thoracic and Vascular Surgery, University Hospital, 53127 Bonn, Germany; jonas.dohmen@ukbonn.de (J.D.); alexander.semaan@ukbonn.de (A.S.); vittorio.branchi@ukbonn.de (V.B.); kalff@uni-bonn.de (J.C.K.); hanno.matthaei@ukbonn.de (H.M.); philipp.lingohr@ukbonn.de (P.L.); 2Institute of Clinical Chemistry and Clinical Pharmacology, University Hospital, 53127 Bonn, Germany; makbule.kobilay@ukbonn.de (M.K.); martinzaleski@gmx.de (M.Z.); gunther.hartmann@ukbonn.de (G.H.); 3QuoData GmbH-Quality & Statistics, 01309 Dresden, Germany; anja.schlierf@quodata.de (A.S.); hettwer@quodata.de (K.H.); uhlig@quodata.de (S.U.); 4CEBIO GmbH—Center for Evaluation of Biomarkers, 81679 Munich, Germany; 5Center for Integrated Oncology (CIO) Cologne/Bonn, 53127 Bonn, Germany

**Keywords:** exosome, microRNA, colorectal carcinoma, biomarker

## Abstract

Background: Despite the significance of colonoscopy for early diagnosis of colorectal adenocarcinoma (CRC), population-wide screening remains challenging, mainly because of low acceptance rates. Herein, exosomal (exo-miR) and free circulating microRNA (c-miR) may be used as liquid biopsies in CRC to identify individuals at risk. Direct comparison of both compartments has shown inconclusive results, which is why we directly compared a panel of 10 microRNAs in this entity. Methods: Exo-miR and c-miR levels were measured using real-time quantitative PCR after isolation from serum specimens in a cohort of 69 patients. Furthermore, results were compared to established tumor markers CEA and CA 19-9. Results: Direct comparison of exo- and c-miR biopsy results showed significantly higher microRNA levels in the exosomal compartment (*p* < 0.001). Exo-Let7, exo-miR-16 and exo-miR-23 significantly differed between CRC and healthy controls (all *p* < 0.05), while no c-miR showed this potential. Sensitivity and specificity can be further enhanced using combinations of multiple exosomal miRNAs. Conclusions: Exosomal microRNA should be considered as a promising biomarker in CRC for future studies. Nonetheless, results may show interference with common comorbidities, which must be taken into account in future studies.

## 1. Introduction

Nearly 1.2 million patients are newly diagnosed with a colorectal carcinoma (CRC) every year, making this entity one of the most common and best studied cancer types worldwide [1]. Slowly progressing, dysplastic adenomas are the most common precursor lesions of CRC, and are part of the adenoma-carcinoma-sequence that presents with a characteristic accumulation of genetic mutations [2,3]. For early detection of CRC, frequently inexpensive and non-invasive fecal occult blood tests (FOBT) or fecal immunochemical tests (FIT) are used, as they have been shown to improve mortality even if they are clearly limited in sensitivity and specificity [4,5]. In contrast, colonoscopy as the golden standard, shows a high sensitivity for the detection of (pre-) cancerous lesions but remains invasive [6], with acceptance in the population being low [7].

Currently used serum biomarkers, such as carcinoembryonic antigen (CEA) or carbohydrate antigen 19-9 (CA19-9), are not recommended for early diagnosis, but for monitoring therapy response in advanced CRC and for detection of recurrent disease [8,9,10]. Therefore, there is a need for highly accurate detection of CRC using minimal invasive biomarkers [11], although using colonoscopy remains inevitable for diagnosis.

A promising class of new biomarkers are small non-coding microRNAs (miRs). They are known to regulate gene expression on a post-transcriptional level by silencing diverse mRNAs and thereby controlling multiple biological processes [12]. Many studies highlight the prognostic and diagnostic significance of tissue and blood-based miRNAs in CRC [13,14,15,16]. However, published results demonstrate a high variability and sometimes conflicting data for specific miRNAs. Another group of promising biomarkers are extracellular vesicles (EVs), and in particular their smallest members, the exosomes. These small spherical vesicles originate from the multivesicular body in the cytoplasm, and are actively secreted into the extracellular space [17]. Exosomes play a crucial role in cell–cell communication and promote the pathogenesis of neoplastic diseases [18]. Remarkably, miRNAs detected in exosomes circulating in the blood show a striking potential as recurrence marker for CRC [19]. EV- and exosome-associated biomarkers provide some advantages over freely circulating biomarkers, as they are protected within exosomes from degradation by proteases or nucleases as a result of their efficient lipid bilayer [20]. Since they furthermore are produced by a unique mechanism [17], exosomes have been proposed to mirror the pathophysiological state of the releasing cells [21,22,23]. Among other studies, this has been demonstrated in chronic myelogenous leukemia cell lines after curcumin treatment using a selective miR-21 sorting into exosomes [24]. Some studies have confirmed the high potential of exosomal miRNAs in the diagnosis and prognosis of diverse cancers, and also CRC [19,25,26]. However, only a few studies compared directly the diagnostic power of free circulating and exosomal miRNAs [26,27,28]. Therefore, the present study assessed profiles of promising miRNA candidates, side by side in exosomes and in the serum of patients suffering from CRC, as well as of patients with colorectal adenomas and healthy individuals as controls. Furthermore, obtained results were compared with established CRC tumor markers.

## 2. Materials and Methods

### 2.1. Patients

Serum samples from a total of 69 patients and donors were investigated in the present study. Among them were 23 patients with CRC, 20 with adenoma and 26 healthy individuals. In CRC patients, samples were prospectively collected at time prior to any treatment, and in adenoma patients prior to routine screening colonoscopy. Patient details are provided in Table 1.

All adenomas and carcinomas were histologically confirmed by an experienced pathologist blinded to patients’ histories. Histologic classification was performed according to the most recent recommendations by the World Health Organization [29], and the latest TNM classification [30]. Advanced adenomas were defined as adenomatous polyps with a size ≥10 mm, ≥25% of villous features or a high-grade dysplasia [31]. Patients with any hereditary form of colorectal carcinomas, history of familial adenomatous polyposis (FAP), Lynch syndrome, inflammatory bowel disease or second primary malignancies at present or past were excluded from the study.

The study was approved by the Institutional Review Board (IRB) of the University Hospital of Bonn (Number: 021/16) 10.02.2016. Blood samples were collected in donors and in patients who were treated at the University Hospital of Bonn, or at affiliated academic institutions between 2011 and 2016, and after written consent was obtained.

### 2.2. Patients’ Characteristics

The study cohort comprised 69 patients (45 ♂, 24 ♀) with a mean age of 62 years [range: 27–85]. Age and gender were distributed equally in all groups (all *p* > 0.05); the CRC group embodied 23 patients (18 ♂, 5 ♀) with a mean age of 64 years [range: 39–81]. This group included three stage I, four stage IIa, five stage IIIb, two stage IIIc and nine stage IV CRCs. The adenoma group contained 20 individuals (11 ♂, 9 ♀) with a mean age of 64 years [range: 27–79]. Fourteen of the adenomas were described as tubular, one as tubulovillous, two as villous and three as serrated adenomas. None of the adenomas showed a high-grade dysplasia or a size over 1 cm, and none was therefore categorized as advanced adenoma. The control group comprised 26 patients (16 ♂, 10 ♀) with a mean age of 58 years [range: 44–85] (Table 1).

**Table 1 diagnostics-12-01413-t001:** Details of patients included in this study.

Groups	*N* (Total *n* = 69)	Age (Years) (Mean/Range)
Healthy control	26	58 (44–85)
Adenoma	20	64 (27–79)
CRC	23	64 (39–81)
**Gender**		**Percentage**
Female	24	34.8
Male	45	65.2
**UICC Stage**		
1	3	13.1
2	4	17.4
3	7	30.4
4	9	39.1
**T-Stage**		
1	1	4.3
2	3	13.1
3	15	65.2
4	4	17.4
**N-Stage**		
0	8	34.8
1	5	21.7
2	10	43.5
**M-Stage**		
0	14	60.9
1	9	39.1

### 2.3. MicroRNA Selection

From among the numerous target miRNAs, we examined 10 miRNAs that have been reported to be altered in serum or CRC tissue specimens in previous studies: miR-let-7c [32,33], miR-16 [34], miR-19a [19,35], miR-21 [13,25,35,36], miR-23a [13,19,25], miR-29 [37], miR-34a [38,39], miR-92a [13,19,40], miR- 222 [14,41,42] and miR-451 [43,44,45,46] (Appendix A) Snord43 was chosen as an internal reference, as this molecule is known to be widely expressed in healthy individuals and has not been found to be altered in CRC [47,48].

### 2.4. Preparation of Blood Samples

Blood samples were collected and carefully processed within a maximum of 4 h after collection by an experienced team, in order to ensure high quality pre-processing standards. All samples were centrifuged at 3000× *g* for 10 min at room temperature for removal of cells and debris. Supernatants were transferred into fresh cryotubes and stored until analysis at −80 °C or were immediately processed. All samples underwent a visual inspection for hemolysis. Red, discoloured or otherwise contaminated samples were excluded from further examination.

### 2.5. Serum Circulating miRNA (c-miR) Isolation

Total miRNA was extracted from serum using the High Pure miRNA Isolation Kit (Roche Molecular Diagnostics, Mannheim, Germany), following the manufacturer’s protocol. In brief, 400 µL of serum was mixed with binding buffer solution and twice centrifuged at room temperature for 1 min at 13,000× *g*. The lysate was spiked with 5 fmol/µL synthetic Caenorhabditis elegans miRNA 39 (Cel-mir-39, Qiagen GmbH, Hilden, Germany). Subsequently, binding enhancer, washing and elution buffer were added, and samples were centrifuged according to the protocol. MiRNA concentrations were measured in duplicate using an Infinite^®^ 200 PRO series spectral photometer (Tecan Group Ltd, Männedorf, Switzerland) at absorbance levels of 260 und 280 nm. Mean c-miR concentrations were used for all further analyses. 

### 2.6. Exosomal microRNA (exo-miR) Isolation

Total exosomal RNAs were extracted from serum using the exoRNeasy Serum/Plasma Kit (Qiagen GmbH, Hilden, Germany). One milliliter of serum was processed according to the manufacturer’s protocol. In brief, serum was homogenized with QIAzol Lysis reagent Qiagen GmbH, Hilden, Germany), and 5 fmol/µL of Cel-miR-39 “spike-in control” was added for extraction control.

Following phenol/guanidine-based combined lysis, membrane-based affinity binding centrifugation was performed, which is described in detail in the manufacturer’s protocol. Exosome separation was then achieved by a chloroform and ethanol-based centrifugation step. To maximize the RNA amount in the last spin protocol step, the RNeasy MinElute spin columns were washed twice using 14 µL of RNase-free water. Each MiRNA concentration was measured in duplicate using an Infinite^®^ 200 PRO series spectral photometer (Tecan). Mean measurements of both exo-miR concentrations were used for all further analyses. Particle size of exosomal RNA was measured in selected cases. Herein, we used the ZetaView^®^ BASIC NTA-Nanoparticle Tracking Video Microscope (https://www.particle-metrix.de/en/products/zetaview-nanoparticle-tracking, accessed date: 28 April 2022) for size comparison and verification.

### 2.7. Measurement of exo-miR and c-miR Levels Using Real-Time Quantitative PCR

Fifty nanograms of exo-miR RNA and 100 ng of c-miR were reverse transcribed using the miScript II Reverse Transcription Kit (Qiagen GmbH, Hilden, Germany) following the manufacturer’s protocol. The different amounts of total RNA input are based on different volumes between the above-mentioned extraction methods. Samples were incubated for 60 min at 37 °C, followed by 5 min at 95 °C and cooled down to 4 °C. Aliquots were then 1:5 diluted with sterile water and immediately processed.

The rt-qPCR was conducted according to the miScript SYBR Green PCR Kit protocol (Qiagen GmbH, Hilden, Germany). First, a 4-microliter template was combined with 10 µL of QuantiTect SYBR Green PCR MasterMix, 10 µL universal–reverse-Primer, RNAse-free water and miR- Primer Assay (2 µL each). Total mixture volume for rt-qPCR was 20 µL. We used the following predesigned miScript PCR primers (all Qiagen GmbH, Hilden, Germany, Appendix A): Hs_miR-23a_2 miScript Primer Assay (MIMAT0000078: 5′AUCACAUUGCCAGGGAUUUCC, Hs_miR-451_1 miScript Pri-mer Assay (MIMAT0001631: 5′AAACCGUUACCAUUACUGAGUU), Hs_miR-34a*_1 miScript Primer Assay (MIMAT0004557: 5′CAAUCAGCAAGUAUACUGCCCU), Ce_miR-39_1 miScript Primer Assay, (MIMAT0000010: 5′UCACCGGGUGUAAAUCA GCUUG), Hs_miR-92_1 miScript Primer Assay (MIMAT0000092: 5′UAUUGCACUU GUCCCGGCCUGU), Hs_miR-16_1 miScript Primer Assay, MIMAT0000069: 5′UAGCAGCACGUAAAUAUUGGCG), Hs_miR-21_2 miScript Primer Assay (MIMAT0000076: 5′UAGCUUAUCAGACUGAUGUUGA), Hs_let-7c_1 miScript Primer Assay (MIMAT0000064: 5′UGAGGUAGUAGGUUGUAUGGUU), Hs_miR-222_2 miScript Primer Assay (MIMAT0000279: 5′AGCUACAUCUGGCUACUGGGU), Hs_miR-19a_1 miScript Primer Assay (MIMAT0000073: 5′UGUGCAAAUCUAUGC AAAACUGA), Hs_miR-29a_1 miScript Primer Assay (MIMAT0000086: 5’UAGCACCAUCUGAAAUCGGUUA) and Hs_SNORD43_11 miScript Primer Assay (MS00007476: 5′CACAGATGATGAACTTATTGAC).

PCR was performed with a LightCycler^®^ 480 system (Roche Molecular Diagnostics, Mannheim, Germany) using the following temperature profile: 15 min at 95 °C of initial denaturation, followed by 45 cycles of 94 °C for 15 s, 55 °C for 30 s and 70 °C for 30 s.

Cel-miR 39 primer assay was used as a control of reverse transcription and PCR function. In addition, Snord43 was used as a housekeeping gene. The relative expression levels of all miRNAs were determined via a separate standard curve for each miRNA. Results were analyzed using the LightCycler^®^ analysis software, Version 1.5.1.62 (Roche Molecular Diagnostics, Mannheim, Germany). The relative miRNA expression values were normalized to Snord43 and calculated using the 2^−ΔΔCT^ method [49]. In order to align the measured miRNA concentration in the real-time quantitative PCR (rt-qPCR), we doubled the standardized exo-miR values. All samples were analyzed in duplicate, and mean values of both measurements were used for statistical analyses. Relative expression levels of the targeted miRNAs were normalized with the following formula and ln-transformed: $$\frac{2^{\mathrm{Target}\;\mathrm{miRNA}(\mathrm{sample})}}{\frac{2^{\mathrm{Housekeeper}(\mathrm{sample})}}{2^{\mathrm{cel}-\mathrm{mir}39(\mathrm{sample})}}}÷\frac{2^{\mathrm{Target}\;\mathrm{miRNA}(\mathrm{calibrator})}}{\frac{2^{\mathrm{Housekeper}\;(\mathrm{calibrator})}}{2^{\mathrm{cel}-\mathrm{mir}39(\mathrm{calibrator})}}}$$

The median levels of Cel-miR-39 with standard deviations are shown in Appendix A.

### 2.8. Statistical Analyses

Values are expressed as means or medians unless otherwise stated. Statistical analyses were performed with the assistance of QuoData GmbH-Quality & Statistics (Dresden, Germany).

MiRNA quantity and tumor markers were assessed for their potential discrimination between the groups. MiRNA levels and tumor markers were log transformed for variance stabilization. Continuous variables were compared using Fisher’s exact test, Mann–Whitney U or the Kruskal–Wallis test, whereas categorical variables were matched using the χ2 test (chi-square) where appropriate. Outliers were identified using Grubbs’ test (significance level 1%) and then excluded from further analysis. Areas under the curve (AUC) of receiver operating characteristic (ROC) curves were calculated, and sensitivities at a 95% specificity level are shown. *p*-values of <0.05 were considered to be statistically significant. Statistical analyses were performed with SPSS (IBM, Armonk, NY, USA, version 23) and GraphPad Prism 4 (GraphPad Software Inc., San Diego, CA, USA).

## 3. Results

### 3.1. Circulating miRNAs Are Concentrated in Exosomes

Median total miRNA concentrations over all groups were higher in the exosomal compartment as compared to free circulating serum miRNA (medians 1504 ng/100 µL [range 200–4118] vs. 1024 ng/100 µL [range 108–2284], *p* < 0.001) (Figure 1a).

Additionally, a stepwise increase of the median total miRNA concentration from healthy controls (1327 ng/100 µL [range 634–2296]) over adenomas (1525 ng/100 µL [range 488–2208]) to CRC (1977 ng/100 µL [range 200–4118]) in exosome total miRNA was noted. In comparison, this gradual increase was not evident for free circulating miRNA in the serum (healthy controls: 822 ng/100 µL [range 108–1800]; adenomas 1214 ng/100 µL [range 316–2284]; CRC 940 ng/100 µL [range 308–1910]) (Figure 1b,c).

The stepwise increment of the median total miRNA levels in exosomes was statistically significant between the control and the CRC group (*p* = 0.003), showing a trend between adenomas and CRCs (*p* = 0.069), while no difference was detected between adenomas and controls (*p* > 0.05). Concerning serum miRNA, only the increase in median miRNA levels from control to adenomas was significant (*p* = 0.005), while all other comparisons between the disease groups showed no differences.

### 3.2. MiRNA Levels in the Exosome Compartment Differentiate between Disease Groups

Out of all tested exo-miRs, three discriminated statistically between CRC vs. healthy controls, while all others showed no differences. Median normalized miRNA values of Let7, miR-16 and miR-23 were significantly higher in the CRC vs. the control group (exo-Let7: *p* = 0.035; exo-miR-16: *p* = 0.022; exo-miR-23: *p* = 0.039). (Figure 2, Table 2; details shown in Appendix A).

Comparison between the adenoma and the CRC group did not reveal a different expression profile for any exo-miR (all *p* > 0.05), while comparison between healthy controls and the adenoma group displayed different levels for exo-Let7, exo-miR-21, exo-miR-23, exo-miR-29 and exo-miR-222.

The highest power of discrimination between CRC and healthy controls was achieved by exo-miR-16 and exo-miR-23 with an area under the curve (AUC) of the receiver operating characteristic (ROC) curve of each being 0.67, and a sensitivity of 4.3% for 95% specificity (Table 3 and Figure 3). Furthermore, we compared microRNA expressions between healthy patients and non-healthy patients (CRC or adenoma). The highest power of discrimination between non-healthy and healthy controls was achieved by exo-miR-23 with an area under the curve (AUC) of the receiver operating characteristic (ROC) curve of 0.69, and a sensitivity of 34.6% for 95% specificity (Appendix A).

Combinations of exosomal miRNA further enhanced the AUCs: A two-marker panel consisting of exo-miR-16 and exo-miR-34 yielded an AUC of 0.74, while the combination of three markers (exo-miR-16, exo-miR-19 and exo-miR-34) resulted in an AUC of 0.80. Finally, combination of exo-miR-16, exo-miR-19, exo-miR-21, exo-miR-34 and exo-miR-222 improved the AUC to 0.87 (Table 4). 

### 3.3. Free Circulating miRNA Levels in the Serum Have Lower Discriminating Potential

In comparison to exo-miRNAs, none of the tested serum miRNAs distinguished CRC from healthy controls (all > 0.05). However, all serum miRNAs showed different median-normalized microRNA levels when adenomas and controls were compared (Appendix A, details in Appendix A). Additionally, all but two free circulating miRNAs in serum (c-miR-19 and c-miR-222) revealed divergent median levels between the adenoma group and the CRC group.

### 3.4. Tumor Marker

The median CEA level was 0.35 [range: 0–4.08] for healthy controls and 1.1 [range: 0–3.4] for CRC patients. The median CA-19-9 level was 1.7 [range: 0–5.36] for healthy controls and 2.64 [range: 0–5.46] for CRC patients (Appendix A). While CEA was not able to discriminate between the study groups (all *p* > 0.05), CA 19-9 was able to differentiate significantly between healthy and CRC patients (*p* = 0.041).

## 4. Discussion

Various reports highlight the potential relevance of miRNAs for diagnosis and prognosis in CRC, although the results sometimes are contradictory [13,14,15,16]. Some major advantages of using miRNAs as biomarkers are their stability against external influences such as pH-alteration, storage, freezing and thawing, as well as their complex involvement in multiple cellular mechanisms that regulate tumor growth and progression [50,51]. In recent years, many research groups tried to optimize the potential of miRNAs as cancer biomarkers. One concept prioritizes the assessment of miRNAs in exosomes as they are frequently released from cancer cells and may present a compartment for the enrichment of cancer-associated miRNAs. Further, miRNAs are better protected within exosomal structures. Despite this promising background, only a few studies have directly compared exosomal and free circulating miRNAs in CRC thus far [26,27,28]. These studies report to a certain extent conflicting results, and also lack a direct comparison with serum tumor markers such as CA 19-9 and CEA.

In accordance with previous reports, our data show an enrichment of total and single miRNAs in the exosomal compartment as compared with free circulating miRNAs in serum [52,53]. Single miRNAs were found to be enriched with up to a thousand-fold in exosomes (miR-16 and miR-19). These results support the hypothesis that exosomes are actively loaded by their cells of origin [21] and show better protection from degradation [27,54,55]. Further, these observations point to the attractive possibility of using exosomal miRNAs rather than free circulating miRNAs for cancer detection. Moreover, three exo-miRNAs (exo-miR-16, exo-miR-23 and exo-let7) showed different levels between CRC and controls but no c-miRNA in serum. Comparable results for exo-miRNAs have been shown in esophageal squamous cell cancer [53], hepatocellular cancer [56] and other gastrointestinal cancers [54]. Furthermore, a significant secretion of colorectal cancer cell-based miRNA encapsuled in exosomes was reported from two colorectal cancer cell lines (HCT-116 and HT-29-) [57]. Other studies have found superior performance of serum miRNAs in the detection of high-risk adenomas and colorectal cancer [28]. Our finding of increased exo-miR-16 levels is in line with recently published results [58] that showed an up-regulation of miR-16 in cancer tissue that predicted disease-free and overall survival in CRC patients. In contrast, other investigators found lower miRNA concentrations in CRC tissue specimens as compared to controls [59,60].

Downregulation of miR-16 has been identified as an independent prognostic factor for survival, although it was expressed in only two thirds of CRC tissue samples [60]. Similar results were reported by You et al. who provided evidence for a direct impact of miR-16 in the post-transcriptional regulation of KRAS, and the potential role of miR-16 as a tumor suppressor in CRC [61]. This discrepancy may be explained by using different study populations, specimen types as well as methodological differences such as isolation, normalization and quantification.

Our finding of increased exo-miR-23 levels in CRC patients corresponds with other reports on elevated miR-23 levels in cancer tissue [62], serum [62] and exosomes [25] of cancer patients. It has been speculated that high miR-23 levels demonstrate a higher potential for migration and invasion of CRC cells as shown in an animal model and in CRC cell lines [63,64]. 

In contrast to other studies, we report elevated exo-let7 levels in exosomes of CRC patients as compared to healthy controls [65,66]. The let7 miRNA family has been described to play a tumor-suppressive role by targeting oncogenes such as RAS and HMGA2 [67]. Contradictory to these findings, elevated levels have been proposed to facilitate tumor immune evasion via suppression of the adaptive immune responses, and are therefore associated with a worse prognosis in CRC [68]. This corresponds with a selective secretion of let-7 family that was found in a distinct metastatic gastric cancer cell line [69]. Nonetheless, further data are needed in order to elucidate the role of exosomal let7 in CRC and other cancer patients.

Other promising miRNAs that were described to be relevant in CRC, such as miR-21 and miR-34, showed no contrasting levels between CRC patients and control, neither in exosomes nor in serum [25,26,70]. In addition, various levels of exo-miRNAs and c-miRNAs were observed in healthy and benign controls, such as in patients with gastrointestinal adenomas. Conflicting results on this issue in the literature highlight the large variability of miRNA expressions in diverse conditions and the importance of appropriate selection of control cohorts. Further, discrepancies may be explained by distinct study populations, isolation and quantification methods, data normalization and interpretation. In addition, solid tumors themselves comprise a variety of different cell types and interact in a complex and not yet fully understood way with the surrounding tumor environment. This inter- and intra-tumorous variety may at least in part explain the inconsistent and partly contradictory results in the serum/exosomal miRNA levels in different studies. Despite the well characterized “adenoma-carcinoma sequence” [71], approximately 10–15% of sporadic CRCs originate from serrated polyps which have been identified as an alternative key pathway [72] that harbors, e.g., the CpG island methylator phenotype (CIMP), and its close relation to BRAF mutations [73]. Consequently, it can be speculated that these “subtypes” of CRC comprise a different miRNA signature.

Since single biomarkers can only cover a part of the biochemical processes present within a tumor, and therefore will have been limited in terms of sensitivity and specificity, combinations of two or more biomarkers can increase the accuracy for diagnosis and monitoring of tumor disease. When combining several miRNAs in our study, we could improve ROC curves for CRC detection vs. healthy controls considerably, which provides a complete picture of sensitivity and specificity over the whole spectrum of cutoff levels. Although this is a promising way to improve cancer detection, it has to be mentioned that the number of cases and controls in the present study was limited, and those data have to be further investigated by additional studies.

Obviously, this was a pilot study with explorative character that focused on miRNAs that were described to be most relevant in CRC. Comparison of blood and tissue miRNAs was not part of the study. As methods for isolation and quantification of exosomes and miRNA have not been standardized thus far, we used commercially available kits and established our own procedures according to MIQE criteria to obtain reliable results. In order to minimize methodical errors and data misinterpretation, and to meet the highest possible processing standard, our sample processing protocols contained many features for quality control, such as standards and controls in all plates, double measurements and adding of exogenous synthetic miRNA cel-miR 39 in order to control reverse transcription, quantification and normalization to Snord-43. Samples themselves were collected according to SOPs of the Biofluid Biobank of the University Hospital Bonn. We used samples from CRC patients as well as from different control groups in order to see specific differences. Sample collection, handling, miRNA assessment and evaluation were done independently, and biostatistical experts were involved in the independent data analyses and interpretation.

In conclusion, we were able to confirm recent study reports on cancer miRNA enrichment in circulating exosomes. Exosomal levels of miR-16, miR-23 and let-7 were different in patients with colorectal cancer when compared with healthy controls, while no differences were seen for serum miRNA. Applying combinations of miRNA markers further increased sensitivity and specificity for CRC detection. Further prospective studies are warranted in order to validate the explorative results of this pilot study.

## Figures and Tables

**Figure 1 diagnostics-12-01413-f001:**
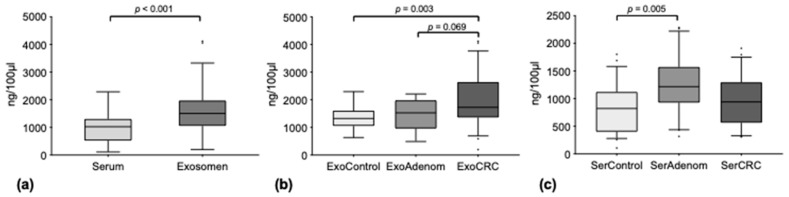
Median total miRNA concentrations (**a**) free circulating in serum and in the exosomal compartment; (**b**) in the exosomal compartment of healthy controls (ExoControl), adenoma patients (ExoAdenoma) and carcinoma patients (ExoCRC); (**c**) free circulating in serum of healthy controls (SerControl), adenoma patients (SerAdenoma) and carcinoma patients (SerCRC). Outliers are plotted as individual dots.

**Figure 2 diagnostics-12-01413-f002:**
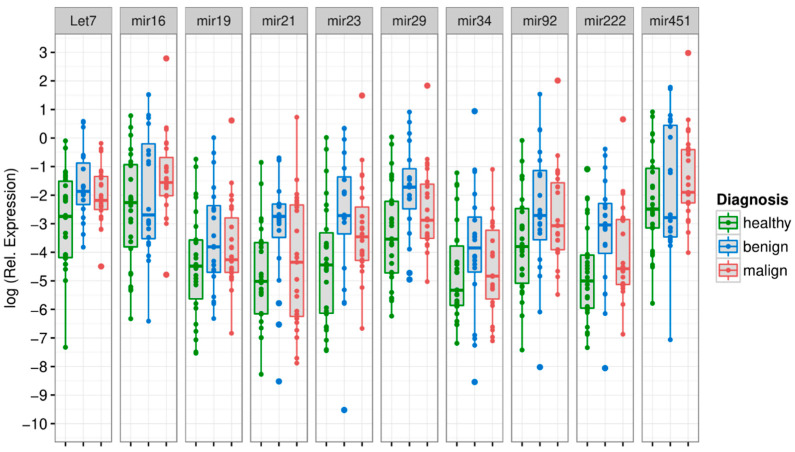
Boxplots of median total miRNA concentrations in the exosomal compartment, differentiated for different diagnoses (healthy, adenoma, carcinoma) and miRNAs. Healthy controls: green boxplot, adenoma patients: blue boxplot, carcinoma patients: red boxplot.

**Figure 3 diagnostics-12-01413-f003:**
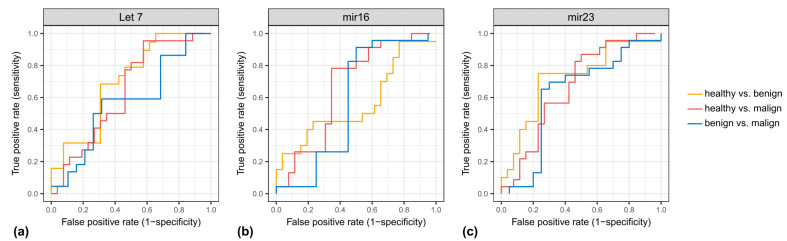
ROC curves of (**a**) Let7, (**b**) mir16 and (**c**) mir23 in the exosomal compartment. X-axis: false positive rate (1-specifity), y-axis: true positive rate (sensitivity).

**Table 2 diagnostics-12-01413-t002:** Median-normalized miRNA values in the exosomal compartment. Comparisons of different diagnoses (healthy controls, adenoma patients, carcinoma patients) were made. Differences are shown as *p*-values. *p*-values of <0.05 (highlighted with *) were considered to be statistically significant.

miRNA	Healthy vs. Adenoma	Healthy vs. CRC	Adenoma vs. CRC
**Let7**	0.008 *	0.035 *	0.333
**mir16**	0.323	0.022 *	0.368
**mir19**	0.098	0.174	0.626
**mir21**	0.006 *	0.444	0.076
**mir23**	0.025 *	0.039 *	0.509
**mir29**	0.005 *	0.079	0.167
**mir34**	0.207	0.783	0.298
**mir92**	0.093	0.063	0.844
**mir222**	0.010 *	0.110	0.213
**mir451**	0.516	0.072	0.466

**Table 3 diagnostics-12-01413-t003:** Areas under the curve (AUC) and sensitivities (SN) at 95% specificity (SP) for the different miRNAs in the exosomal compartment. Comparisons are between healthy and CRC patients.

miRNA	Healthy (neg.) vs. CRC (pos.)	
	AUC	SN@95%SP
**Let7**	0.64	4.5%
**mir16**	0.67	4.3%
**mir19**	0.60	4.3%
**mir21**	0.55	8.7%
**mir23**	0.67	4.3%
**mir29**	0.64	4.3%
**mir34**	0.55	4.5%
**mir92**	0.64	8.7%
**mir222**	0.63	13.0%
**mir451**	0.65	4.3%

**Table 4 diagnostics-12-01413-t004:** AUCs and sensitivities at 95% specificity for different combinations of miRNA in the exosomal compartment.

Combinations	AUC	SN@95%SP
10 miRNAs	0.90	65.2
5 miRNAs: mir16, mir19, mir21, mir34, mir222	0.87	56.5
3 miRNAs: mir16, mir19, mir34	0.80	30.4
2 miRNAs: mir16, mir34	0.74	21.7

## Data Availability

The data presented in this study are available on request from the corresponding author. The data are not publicly available due to privacy restrictions.

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
