# Peer review of "Diagnostic Potential of Exosomal microRNAs in Colorectal Cancer"

_diagnostics, 2022, doi:10.3390/diagnostics12061413_

Round 1

Reviewer 1 Report

 In this work, Dohmen and co-authors assessed differences in expression of well-known microRNAs, extensively reported in literature, analyzing serum samples from a small cohort of healthy donors and patients with either adenoma or colorectal cancer. It is opinion of this reviewer that this study adds very little novelty in the field and raises several concerns that should be carefully addressed.

Major Revision:

  • Authors report Ref. 39 to justify the choice of miR-34a. However, Ref.39 analyzes miR-34a-5p, while Dohmen and colleague analyzes miR-34a-3p (line 152). May want to discuss why they chose to analyze the miR star strand instead of the main one and report adequate literature in support of that choice.
  • In the introduction, Authors aim to investigate if microRNAs could be used as biomarkers to identify individuals at risk of CRC. To do that, they should also compare microRNA expression between patients with adenoma/CRC and healthy donors.
  • It is opinion of this reviewer that quality controls are missing. What kind of controls have been adopted to check if their samples were not affected by hemolysis? After having performed the exosomes extraction with the commercial kit, did Authors assess the diameter of microvesicles (e.g., nanosight analysis) to make sure they are analyzing only the exosomal fraction? In supplementary figure, cel-miR-39 RT-qPCT data could be also reported to make sure that spike-in levels are identical among samples.

Minor Revision:

  •  In general, Authors should carefully read the references they are going to report. For example, in line 284, HCT-116-379 and -NTC are reported as if they were two different cell lines instead of the same cell lines overexpressing or not miR-379.
  •  A careful revision of the manuscript should be performed to avoid some typos (e.g., in line 118 puffer instead of buffer).
  • In general, some references and guidelines are dated. Authors should report more recent references in order to provide to readers the latest knowledge on this field.

Reviewer 2 Report

The manuscript is properly prepared and edited. The research was carried out correctly and the results were analyzed using well-chosen statistical tests. Figures are aesthetic and legible. The conclusions are supported by the obtained and described results, hence the content does not raise any objections. The biggest disadvantage of the work is the small size of the compared groups.

In my opinion, the introduction should include a table with the characteristics of the miRs analyzed in the study (e.g. biological role, target genes, correlation of changes in the level of expression of a given molecule with the pathological process). Similarly, in the methodology, it is worth preparing a table for the characteristics of primers. Please also correct the citations in the text 

Round 2

Reviewer 1 Report

It is opinion of this reviewer that the manuscript of Dohmen and coworkers still presents methodological flaws.

As extensively reported in literature (e.g., https://doi.org/10.1016/j.mam.2019.10.005), hemolysis may seriously alter miRNA evaluation in liquid biopsies and visual inspection is not a sufficient quality control. The fact that the Authors cannot exclude the possibility of PBMC contamination in their samples impacts on the reliability of their results.

Moreover, answering to my second point, the Authors claims that they cannot exclude that their controls are truly healthy since they could have adenomas too.

I believe that a rigorous selection of patients and methodological workflow are mandatory prerequisites for this type of studies. Without these, I could not trust any results coming from this study. 

Minor point: I still do not understand why the Authors choose to "consciously use the lesser-known star strand of miR-34a-3p" instead of the more "commonly and widely examined" miR-34a-5p, if their study is supposed to assess profiles of promising miRNA candidates (as stated in lines 66-67).